# Social Justice and Inclusive Education in Holy Cross Education in Bangladesh: The Case of Notre Dame College

Md Shaikh Farid 

Department of World Religions and Culture, University of Dhaka, Dhaka 1000, Bangladesh; sfarid@du.ac.bd

**Abstract:** This paper examines how Holy Cross missionaries in Bangladesh have interpreted the Catholic Church's teachings on social justice and inclusive education and have implemented its recommendations at Notre Dame College. The Catholic Church's documents on education provide direction, purpose, and rationale for Catholics across the globe. These documents advocate Catholic educators toward social justice in education by making education available, accessible, and affordable to all. This leads to the question of how Holy Cross adopts social justice and inclusive education at its elite educational institutions such as NDC, which charges high tuition and enrolls mostly urban meritorious students. The paper is based primarily on a combination of the examination of written documents and fieldwork involving interviews with Holy Cross personnel. The study reveals that the Catholic concept of social justice, social teachings and inclusive education are applied partially at NDC. As recommended by the Catholic Church, Holy Cross educators have taken different educational programs and social projects—both formal and non-formal—to serve the poor and underprivileged at Notre Dame College. However, as the admission policy of the college is based on the results of previous examinations, there is very little scope for the poor and underprivileged groups to get admitted to the college. Furthermore, the institution fails to include children with special educational needs because there are no special opportunities at the college for students with special educational needs.

**Keywords:** Holy Cross; missionary education; catholic education; social justice; inclusive education

## 1. Introduction

The Congregation of Holy Cross, a Catholic religious congregation, has been working on evangelization and education in Bangladesh since 1853. The Catholic Church's documents on education provide direction, purpose, and rationale for school administrators, principals, teachers, and leaders in different countries across the globe. The documents are also considered to provide authoritative guidance on the nature and purpose of a Catholic education worldwide, and how Catholic education is or ought to be. The Catholic Church's documents on education provide an overview of ways in which Catholic schools can foster social justice in education (McKinney 2018). As recommended by the Catholic Church, Holy Cross educators have taken different educational programs to serve the poor and underprivileged in Bangladesh. For example, Dziekan (2002) mentioned that Holy Cross schools and colleges find ways to provide educational opportunities for the poor and disadvantaged groups as a part of social justice. Timm (2000) argued that the Congregation of Holy Cross has made a preferential option for the poor. Thus, all the schools have some provisions for helping poor children to attend Catholic schools. Morin (1994) argued that, for apostolic works, the tribal people, the poor, and the marginalized are their priority. Peixotto (1992c) claimed that one of the primary goals of the Church is to serve the poor and oppressed. He also mentioned that Notre Dame College was founded specially to give attention to the minorities, poor, tribal, and other underprivileged groups. Thus, Notre Dame College has been operating a literacy school for the slum dwellers and street children and has special programs for the poor and underprivileged. However, the concept of social

justice and inclusive education have become problematic as questions arise about how these principles are being practiced in Catholic schools around the world that charge tuition and educate wealthy students (Grace and O'Keefe 2007). Grace (2002) argued that Catholic schools are in a difficult position as the government policies of the market economy in education sectors. As a result, Catholic education is adopting a policy of giving more priority to academic achievement sacrificing inclusive education and option for the poor. This leads to the question of how Holy Cross adopts social justice and inclusive education in its elite educational institutions that charge high tuition and enroll mostly urban meritorious students. This paper investigates how Holy Cross missionaries in Bangladesh interpret the Catholic Church's concept of social justice and implement its recommendations on social justice issues at Notre Dame College.

Notre Dame College is the first Catholic college founded by the Congregation of Holy Cross. The college was established in *Lakshmibazar*, Dhaka in November 1949. In the beginning, the college was named after St. Gregory and was started under John Harrington, the first principle of the college (Gillespie and Peixotto 2001). From the beginning of the college, the classes were held in St. Gregory's High School. In 1954, the college was moved to its present-day location in Matijeel, Dhaka, and was renamed Notre Dame College. The name was given as a tribute to the University of Notre Dame, the alma mater of many of the faculty members. The college was opened with two sections—Arts and Commerce—with nineteen students. In 1952, the college began its B.A. (pass course); and in 1953, the college opened its science section. It expanded over the years, and currently, there are 3000 students in the Class XI and XII, and 120 in B.A. In the beginning, the college was started as an English medium college and was mostly taught by American Holy Cross priests. However, after independence, the language of instruction was changed from English to Bengali. Within a few years after its establishment, the college became one of the best colleges in Bangladesh (then East Pakistan). Even today, the college is considered as one of the best colleges in the country. From its inception, the college had two main goals: to give college-level education to Christian students and to provide quality and value-based education to students to contribute to the development of the country. In keeping with social justice ideas, special attention is given on students who would otherwise be denied the opportunity for such a quality and value-based education due to their economic circumstances, ethnicity, and rurality (Farid 2020; Gillespie and Peixotto 2001).

## 2. Literature Review

### 2.1. Social Justice in Catholic Education

The term "social justice" refers to the principles of equality and equity in all aspects of life for all members of a community. These principles are based on perspectives and consistent practices that are perpetually all-inclusive for daily life and the way human beings should live (Calley et al. 2011). By the same token, these orientations are opposed to any measures, activities, or behaviors that lead to any kind of partiality in one's life in relation to others. Mantras such as "Of all at all times," "For all in all places," and "Never willfully against anybody" are given priority in matters of social justice (Boakari 2010). In other words, social justice is an umbrella concept that attempts to explain and describe the fundamental principles of equality, equity, respect for other people's dignity, and respect for the environment (Valadez 2015). Social justice encompasses not only what individuals can achieve on their own, but also how they are affected, assisted, or hindered by institutions (Byron 1998; Heft 2006). These ideals, with the goal of ensuring human dignity for individuals, have been enshrined in important historical documents, such as the Universal Declaration of Human Rights of the United Nations (Gewirtz 2006; Sturman 1997).

The Catholic Church sees social justice as a mission for all men, women, youth, and children, and understands the necessity of social justice and living according to the principles of justice as human beings with/to other humans (Boff 2012). Social justice and its virtues—avoiding racism and classism, eradicating discrimination, fighting for gender equality, living for peace, overcoming prejudice, standing in solidarity with the poor, living

a life of stewardship, and teaching tolerance—are external practices that emerge from an interior reality of love (Canales and Min 2010).

Catholic social teaching is an essential aspect of the Catholic intellectual tradition (Heft 2006). Commitment to social justice is an essential character of Catholic schools. According to *The Catholic School* (Congregation for Catholic Education 1977), the schools promote this purpose in two ways:

> *Since it is motivated by the Christian ideal, the Catholic school is particularly sensitive to the call from every part of the world for a more just society, and it tries to make its own contribution towards it. It does not stop at courageous teaching of the demands of justice even in the face of local opposition, but tries to put these demands into practice in its own community in the daily life of the school.* (p. 58)

The Catholic school teaches social justice and justice as a part of the worldview shaped by the curriculum. Projects, such as Christian ministries, are also a means of teaching the students social justice. If anything sets them apart, students who graduate from Catholic schools should be marked by a passion for social justice. The Catholic school's Catholic nature is evidently more than its role in faith formation. Catholic schools are Church-run organizations dedicated to upholding and promoting human dignity (Muderedzwa 2022). They do so by providing an all-encompassing education, stimulating the development of the human community, advocating for the abolition of all forms of oppression, and promoting cultural progress (Youniss et al. 2000).

Social justice draws on several essential notions from Catholic social teaching, such as the common good and human dignity. Social injustice is also known as social sin, which is the result of individuals' misdeeds compounding to create unjust societal conditions. Many political philosophers today, following John Rawls, see justice as the primary virtue of social institutions, whereas the Greeks saw it as an ethical virtue practiced by individuals. To put it another way, justice is inherently social (Heft 2006).

As one of their primary ways of acting in a socially just manner, Catholic schools identify ways to assist the needy. Catholic schools participate in the Church's mission to evangelize. To meet this responsibility, the Church has the right to establish and direct schools for any subject, type, or degree. In fact, she founds her own schools because she sees them as a preferred means of promoting the education of the whole person since the school is a center in which a specific view of the world, people and history is developed and conveyed (Miller 2007).

### 2.2. Inclusive Education in Catholic Education

Education is a fundamental human right. It means that everyone, including persons with disabilities, both men and women, has the right to a good education. Inclusive education is the heart of the human right to education. As a natural consequence of this right, all children have the right to an education that is free of discrimination based on disability, ethnicity, religion, language, gender, capabilities, culture, and other factors (Rodriguez and Garro-Gil 2015). Inclusive education is defined as a strategy for transforming the educational system by removing barriers that prevent all students from participating fully in schools. Inclusive education seeks to: (1) provide all learners with the greatest possible chance to obtain quality education in accordance with their needs and abilities; (2) actualize the implementation of education that respects diversity and is non-discriminatory for all learners (Wahyuningsih 2016).

From Catholic social teaching to the example of Jesus as a teacher in the scriptures, the basis for inclusion is clear in Catholic education (Bonfiglio and Kroh 2020). Catholic means inclusive/universal that includes welcoming all, and embracing diverse others in a participatory and connected community. Catholicism urges people to appreciate both the unity and diversity of life and to feel God's spirit as love that permeates all of creation (Groome 1996). This effort is done in the name of Christ, who is at the center of the Catholic school's mission. The Catholic school seeks to create a school community that is driven and based on the gospel spirit (Pontifical Council for Promoting New Evangelization 2020). As

a result, Catholic schools model themselves after the demands of the gospel and Christian living (McKinney and Hill 2010). Therefore, the Catholic Church allows Christians and non-Christians to study in Catholic schools. In *Declaration on Christian Education* (Congregation for Catholic Education 1965), the Church stated:

> *The Church considers very dear to her heart those Catholic schools . . . which are attended also by students who are not Catholics . . . This Sacred Council of the Church earnestly entreats pastors and all the faithful to spare no sacrifice in helping Catholic schools fulfill this function . . . especially in caring for the needs of those . . . who are strangers to the gift of faith.* (§9)

*The Catholic School* states, "the Catholic school offers itself to all, non-Christians included, with all its distinctive aims and means, acknowledging, preserving and promoting the spiritual and moral qualities, the social and cultural values, which characterize different civilizations" (para. 85). Similarly, *The Catholic School on the Threshold of the Third Millennium* (Congregation for Catholic Education 1997), emphasizes "[Catholic education] is not reserved to Catholics only, but is open to all those who appreciate and share its qualified educational project" (p. 16).

One of the distinctive aspects of Catholic education is that it is open to everybody, particularly the poor and weakest in society (Tete 2007). According to the Holy See, every Catholic school's mission must include assistance to the needy. Education is a key factor in improving the social and economic condition of disadvantaged individuals and people. Church schools must take the appropriate steps to ensure that minority and underprivileged students are included in their student body. They should also pay extra attention to students who have specific needs, whether those needs are caused by natural deficiencies or family problems. Catholic schools must continue to demonstrate Christ's love for the poor in order to be true to their mission of giving to all, especially the poor and marginalized (Ruankool 2022). The Apostolic See is committed to supporting all efforts to make Catholic education affordable and accessible to all girls and boys (Miller 2007). *The Catholic School* states:

> *First and foremost the Church offers its educational service to the poor, or those who are deprived of family help and affection or those who are far from the faith. Since education is an important means of improving the social and economic condition of the individual and of peoples, if the Catholic school was to turn attention exclusively or predominantly to those from wealthier social classes it could be contributing towards maintaining their privileged position and could thereby continue to favour a society which is unjust.* (pp. 44–45)

Catholic schools educate the poor and underprivileged groups because serving the poor is an essential element of the mission of Catholic schools and the Catholic Church (Wodon 2019a). The preferential option of the poor has long been a prime agenda of Catholic social teaching and the principle also applies to the mission of Catholic schools (McKinney 2021). The Congregation for Catholic Education warned that to be financially self-supporting, Catholic schools would be admitting children from weather families, while the Church should give its educational services to the poor instead (Congregation for Catholic Education 1977).

Studies reveal that Catholic schools do not always favor the underprivileged. This is not surprising as it is difficult to reach the poor because cost recovery makes Catholic schools' services less accessible to the poor in the absence of or with little government assistance. As a result, while Catholic and denominational schools have a greater ability to serve the underprivileged than private secular schools, they generally have fewer underprivileged pupils than public schools (Wodon 2019a, 2019b). Grace and O'Keefe (2007) noted that Catholic schools' ability to serve the poor have been declining in many countries. Catholic schools are primarily located in low-income countries. Studies suggest that although Catholic schools in Africa serve many children in poverty, they proportionately serve more children from well-off families. Therefore, the presence of schools in low-income

nations does not ensure that Catholic schools serve underprivileged children (Muderedzwa 2022; Wodon 2019a).

However, many Catholic institutions continue to serve the wealthier socioeconomic classes. The reasons for this are complicated, encompassing not only some schooling systems' inherent conservatism, but also genuine financial limits in servicing the poor if governmental aid or subsidy is not available. There are tremendous temptations for Catholic schools to admit students who will "add value" to the school's reputation and image in nations where visible academic results published in league tables and amplified by the media are particularly important. This could mean that children from lower socioeconomic and cultural backgrounds do not have the same access to the best Catholic schools compare to students from higher socioeconomic backgrounds. The extent to which a competitive market mentality has infiltrated the world of Catholic education poses a serious challenge to the institutions' mission integrity (Tete 2007).

Similar to the preferential option for the poor, Church documents support the idea of including people with disabilities in Catholic education (Frabutt et al. 2013). The tenets of the Catholic faith include the idea of including students with disabilities in Catholic schools (Boyle 2020). Despite the appeal, Catholic schools have a long history of underserving diverse populations, particularly students with disabilities (SWDs). In reality, while Catholic schools try to serve all students, they can fall short of this ideal for a variety of reasons (Scanlan 2009). Despite the evidence supporting inclusion, certain barriers have been identified that present difficulties for schools in implementing SWDs integration, including: (a) lack of culture, (b) lack of resources, and (c) lack of knowledge and capabilities. Negative teacher attitudes have emerged as one of the major barriers to inclusion (Crockett 2012). A teacher may feel that students with disabilities need a specialized environment to be successful. Such negativity spoils the culture and prevents a shared philosophy of 'all are welcome', which is necessary to support students at different levels of ability (Causton-Theoharis et al. 2011).

Meaningful inclusion requires significant resources. Not surprisingly, incorporating SWDs into mainstream education may require additional faculty and staff (i.e., special education teachers, paraprofessionals, and specialists). Additionally, alternative curricula, activities, and technologies may be required to support different learning needs. For these reasons, many view inclusion as a costly endeavor (Bonfiglio and Kroh 2020; Crockett 2012). Catholic schools receive a small grant from governments to provide services to students with identified disabilities. In the past, the lack of funding in Catholic schools has been cited as a barrier to inclusion (Crowley and Wall 2007). Another obstacle to the implementation of inclusion in Catholic schools is a lack of staff knowledge and skills. A lack of specialized education and experience as an educator has been reported to be a barrier to the successful care of students with disabilities (Durow 2007).

In *All Are Welcome: Catholic Schools' Inclusive Service Delivery*, Martin Scanlan (2009) situates this contradiction and urges Catholic schools to review how they are fulfilling the needs of different learners, particularly special needs pupils and English language learners. He argues that Catholic schools are called to serve the impoverished and downtrodden as part of their devotion to Catholic Social Teaching (CST). The fundamental essence of a Catholic school, according to Catholic social teaching, is its dedication to assisting those who have traditionally been marginalized. All people have inherent dignity, goodness, and Godliness, according to CST. Catholic schools must endeavor to not only incorporate CST into the curriculum, but also to use CST as a framework for structuring their schools, particularly in terms of attracting and maintaining students from various backgrounds and with special needs (Popper 2011).

Many studies demonstrate that Catholic schools enjoy a very good reputation, but it comes at a greater price, making it challenging for them to serve the most disadvantaged. Even though it is not wise to draw general conclusions from such anecdotal data, it is widely believed in many nations that Catholic and more broadly Christian schools offer high-quality education, possibly at the sacrifice of their capacity to serve more low-income

children (Bryk et al. 1993; Grace 2003; O'Keefe and Scheopner 2009). One of the difficult challenges that Catholic schools face relating to two missions that are not always easy to reconcile: providing a quality education to students with affordable education for the poor (Wodon 2019a). Despite these limitations, it is clear that Catholic schools are succeeding in reaching millions of children living in poverty (Wodon 2020). The majority of Catholic schools are for the poor and underprivileged. Great importance is attached to educating the students according to national requirements. However, locally available resources are not sufficient to ensure sustainable and quality education, especially in rural schools (Rasiah 2020).

## 3. Methodology

The research is based primarily on a combination of examining written documents and field work involving interviews. A qualitative case study research method of investigation is chosen. The available published and unpublished documents, especially in Bangladesh regarding the Holy Cross education were critically and objectively examined. I get access to published and unpublished sources in Holy Cross Provincial Archives of Priests, Brothers, and Sisters, Notre Dame College Archive, St. Gregory's High School and College.

As well as interviews of the locally available Holy Cross missionaries—local and foreigners—including priests and brothers, who were/are involved in both teaching and administration at Notre Dame College were conducted. Fourteen participants were interviewed using a semi-structured interview. Four of them were American Holy Cross missionaries who worked at the college as principals. Open-ended questions were posed, and an unstructured interviewing format was adopted. This allowed the research participants to express their ideas using their own words instead of having to fit their knowledge into pre-set questions determined by the researcher.

Two important techniques of the interview were selected. First, the length of the interviews was not be predetermined; rather, each interview was sufficiently long for the establishment of rapport between the two parties. Second, there was no dependency on a list of questions; instead, initial questions were based on the guiding questions. The nature of the response provided the direction of interviews. Interviews were tape-recorded with the prior-permission of participants. Informed consent was obtained from participants. Complete information was provided to them about the research. Participants' anonymity, privacy, and confidentiality were ensured, but almost all participants expressed their willingness to disclose their identity as they find it non-problematic.

Data derived from different sources were analyzed together, to identify similarities and themes. After the transcription of the interviews, the transcriptions were examined thoroughly, and the key points of each interview from those transcripts, together with data from documents, were sorted out into key themes, patterns, and categories. The data analysis was guided by Miles and Huberman's (1994) technique of data analysis.

The case study methodology is employed in this research to gain an in-depth understanding of the case studied, and the meaning involved therein. The study includes Notre Dame College as a case study institution. Since Holy Cross consists of three different branches—priests, brothers, and sisters—each branch has its educational institutions, while they all belong to Holy Cross and are centrally administered. The paper selects an elite institution that belongs to the Priests. The college is located in Dhaka, the capital city of Bangladesh.

## 4. Findings and Discussion

This section of the research presents the findings of an exploratory study of a Holy Cross elite institution in Dhaka, which focuses on a Christian educational tradition operating in a primarily non-Christian country. According to the interviewees of the study, key characteristics of Holy Cross schooling in Bangladesh are based on Christian values. These values are being implemented in different ways, including providing education to their communities; giving access to education to the poor and underprivileged; charitable and social involvement and activities; concern, and solidarity among teachers and

within the school, and creating a friendly and family-like school environment; providing opportunities for religious practices for all students belonging to all religious traditions; developing cultural and social awareness, societal betterment and transformation; and responsiveness to the disadvantaged. These exercises encouraged openness, cooperation, care, and awareness. However, these are not a part of the curricula and daily life at the college. Non-formal and extracurricular activities serve as the main means of addressing this. The following sub-sections present and discuss the findings of the study.

*4.1. Educational Options for the Poor at NDC*

The college has different educational programs both formal and non-formal to serve the poor and underprivileged groups in Bangladesh. Dziekan (2002) mentioned that the college finds ways to provide educational opportunities for the poor and disadvantaged even at the cost of considerable sacrifice that requires the provision of financial assistance to the students who cannot afford full tuition and the provision of educational support programs for those who require special assistance with learning. Peixotto (1992b) claims that one of the primary goals of the college is to serve the poor and underprivileged groups. Richard Timm, former principal of Notre Dame College, mentions that they are trying to bring up the good citizens, and the people who are well-formed in relationships to the poor (Interview with Timm). Similarly, David Burrell argues that they give importance to the formation of human values, respect for persons, and dignity of the human person, and attention to the poor and underprivileged (Interview with Burrell). Similarly, Benjamin Costa, notes that they give attention to the poor and underprivileged. He claims that there are many institutions in Dhaka, but no institutions are helping the poor. They do it; others don't do it. They have many slum children in their colleges. He said, "We do it because my faith tells me; my faith compels me to that" (Interview with Costa). He claims that Notre Dame College was founded specially to give attention to the minorities, poor, tribal, and those who are underprivileged groups to educate them and provide service to them.

Joseph Peixotto, former principal of Notre Dame College, also mentioned that although Notre Dame College was founded to serve the Christian community and the country, after the independence the college involved more in social works. The Church emphasized, "preferential option for the poor" in the new country that was such a great need; therefore, they reflected on how their institutions such as Notre Dame College can attach more to that program. As a result, they made some changes to college policies to bring some more students and open programs for poor students mainly from villages (Interview with Peixotto). The college decided that one of its main goals must be to try to inspire students to become socially conscious. Thus, the college started several social projects on campus. To the extent possible it tries to get students involved in social projects, or at least to make them aware of what the college tries to do in order that they might appreciate the values the college considers as primary (Peixotto 1992a). Timm mentioned "We are trying to bring up the good citizens, to bring up the people who form this country, well formed in relationship to the poor" (Interview with Timm). Similarly, Peixotto (1992a) notes:

> *From the time of the foundation of Bangladesh, the college has made a determined effort to operate projects for the poor. These projects not only provide direct service to the poor but also proclaim to students the priorities of the college itself—with the hope that these future leaders of society will become imbued with the same spirit of social concern and service to the poor and oppressed. The college's efforts are in line with the priority of the church of Bangladesh of giving the preferential option to the poor.* (p. 7)

Timm (2002) claims that education for the poor is a basic concern of Holy Cross. From the time of the foundation of Bangladesh, the college has made a determined effort to operate for the poor. For example, he mentions that since 1972 Notre Dame College has been operating a literacy school for slum dwellers and street children. The college was one of the first institutions in this new nation to respond to the appeal by the President to provide literacy instruction (Gillespie and Peixotto 2001). The priests at NDC felt that something should be done for the poor and slum dwellers, and one task would be to offer

literacy training for the masses. The principal of NDC hired a couple of teachers and started school with 40 children. Over the years, the total enrolment has grown to 1200. Thus, the school has to have three shifts: morning, afternoon, and evening. The night shift of the literacy school is for children and adults living in slums, most of whom work during the day. They come to study in classes from kindergarten to class seven (Peixotto 1992a, 1992b, 1992c). These classes are taught by the Martin Hall students, who are poor students on a special work program that enable them to pay their college fees and thereby study at the college for free (Timm 2002). Of all the social projects conducted at NDC, the school seems to have the widest impact and must therefore continue to be given top priority (Banas 1999). The college has also a trade school for poor students, there is a dispensary and sick shelter, and handicraft work for destitute women (Peixotto 1992a, 1992b, 1992c).

The student work program is Notre Dame's main means of making education available to students from poor families, particularly for Christian students. The college started the program in 1976. Each year the director of the program sends letters to pastors and headmasters of Catholic schools asking for names of students who are completing their high school studies, are capable of studying in the college, but who cannot afford the cost of college studies. After examination of the requirements for admission to the college, about 50 students are selected each year. They live in Martin Hall, do part-time work at the college, and carry on their two-year intermediate college course of study. They work on the college grounds, offices, labs, and serve as gatekeepers and so on (Peixotto 1992a, 1992b, 1992c). Frank Qunilivan mentioned:

> *You know the College has Martin Hall, which has 140 boys with the capacity from all over Bangladesh. you know a fair number are tribal kids from local area who get a change to be at NDC to study there. We say, regular student body, we take the best students of the country and two year later they remain best students of the county that is not the thing we want, is not appreciable. We take kids from Bandarbon, Thakhurgoan, every year and every one of those kids pass and pass decently. NDC has been conscious of the poor students who are meritorious.* (Interview with Qunilivan)

Timm, a former principal of the college, claims that the student work program is highly beneficial both for the students and the college itself. He mentioned:

> *The program provides many benefits. Students who otherwise could not possibly study in the college get an education with regular classes and very strictly supervised study conditions. Many gather valuable work experiences. Those who earn good reputations get good jobs. Seldom in the society can an employer find a person who has completed his studies while at the same time does not hesitate to dirty his hands with whatever job is needed to be done. The college is greatly benefited because many jobs are done and done well thanks to this large workforce. Some of these students return to their rural parishes and contribute to the work of the church and their communities.* (Interview with Timm)

The program has enabled many tribal students to study at NDC. Although most students in the program are Christians, a small number are Muslims, Hindus, or Buddhists (Gillespie and Peixotto 2001). Peixotto mentioned how and why the college started the student work program at the college:

> *We developed the system over the years, particularly after the independence. We made very special effort the serve the people who don't have the opportunity other institutions, we did not want to be like other institutions. We wanted to be in supplements, we have fancy colleges for rich people, we have government colleges where you no need to pay fees at all. We cannot do that, but we have a work-student program, those who qualify they can work and pay the tuition.* (Interview with Peixotto)

The social work programs of the case study institution started as a result of Vatican II's recommendation. They got more involved in such social activities because of self-criticism, and a recommendation of the Bishop's Institute for Social Action held at Kuala Lumpur in 1975 (Gillespie and Peixotto 2001). The conference was very critical of leading urban

Catholic institutions as being elitist and a part of the oppressive, unjust structure of society. The HC priests at the conference considered these criticisms thoroughly in their apostolic plan meeting for the District Chapter of 1978. The priests raised the issue again in their apostolic plan for the District Chapter of 1981 (Apostolic Plan for Notre Dame College 1976). They concluded that NDC was indeed elitist, but not as much as many Catholic institutions of India, and they took steps to be oriented towards the poor in Bangladesh (Gillespie and Peixotto 2001). Thus, they took a reformist position in seeking ways to make those institutions centers for awareness building about the structures of society that work against the poor and powerless.

Although they had been emphasized on 'preferential option for the poor' in education, they intensified the program in post-war Bangladesh. They emphasized "preferential option for the poor" in the new country because there was a great need to support the people of the country, and they reflected on how their institutions such as Notre Dame College can attach more to that program. Therefore, they made some changes to college policies to bring some more students and open programs for poor students mainly from villages. Thus, the college contributed to the application of the ideal of the Church in society (Interview with Peixotto).

Although the HC missionaries claimed that NDC had been serving the poor and underprivileged through formal and non-formal educational programs, the poor students who got financial support or who were admitted under the Martin Hall programs and related programs belonged to the Catholic community. The poor and underprivileged belonging to other religions were mostly excluded from this program.

### 4.2. Catholic Principle of Social Justice in Practice at NDC: Rhetoric and Reality

When asked how effectively Holy Cross case study institute upholds Catholic education principles, such as social justice, Catholic social teaching, the responses were mixed. Many view that the college is very effective in discipline, but not very good in values such as humanity, social interaction, human dignity, helping others in need, and respect for others because the current education system does not give them these opportunities. Moreover, it mostly depends on the teachers, and if teachers do not have such intention, they merely give them what is prescribed by the curriculum and written in text books.

Respondents said that the college sees social justice discourse the least, suggesting that social justice is not seen as fundamental or important in the educational offering. Social justice does not seem to be as significant in the case study institution as seeing education as a tool for employability and economic development. When asked the question: 'How do you address issues like social justice, human rights in the college?' Many of them replied that while they could not discuss the subject with their students, they could still uphold human rights in the classroom by making an effort to treat all students with respect in interactions and throughout their college life. When asked, 'What Catholic principles guide Notre Dame College?' In response, many of them said that they do give marginalized pupils extra attention and give them an opportunity to study in the college, have some rural and minority quota, follow up with the students, care for the marginalized, love for the impoverished, for one another, shoulder one another's problems.

Principals at the case study college stressed the need of upholding strict academic standards, and they frequently turned away many applicants who lacked the necessary preparation or performance. They claim that in order for the college to remain competitive and uphold a respectable nationwide, the institution's reputation is important to the academic achievement of its pupils. The principal of the college said that academic success is the most common way of assessing a college's worth and prestige in Bangladesh, so success is more attached to the institution than to the students. According to principal, his college enjoys a good name in Dhaka and is well seen and esteemed because its students would easily go to reputed universities, such as BUET, medical colleges, and Dhaka University for higher studies (Interview with Hemanto).

Another former principal remarked on the elitist reputation of the college, noting that it is very challenging for them to adapt their methods now that they are well-known and people think that is the place where you actually learn things (Interview with Costa). Most of them acknowledge that while their ability to choose the students they want to give them an advantage, doing so is not particularly honorable of them because they only keep the brightest students and ignore the others.

Although Holy Cross case study institution is very clear about its missions, the degree to which they actually address its missions and identity in practice falls short. There is a sizable gap between intention and reality in this case. Although Holy Cross Congregation in Bangladesh provides educational opportunities for the poor, the case study institution is among the most elite and narrowly selective in the city, providing a privileged option to the wealthy and well-connected. Although the participants in the study emphasized the humanitarian message of Holy Cross education, in practice the case study elite college places a strong focus on academic results and excellence. Moreover, the college mostly fails to incorporate students with disabilities as there is no provision in the college to give access to the students with special educational needs.

Despite some limitations in implementing the Catholic idea of social justice in education, Holy Cross Congregation is one of the few education providers in Dhaka that teaches their students about human value and dignity, as well as a humanitarian approach to everyday life. While the Congregation of Holy Cross provides quality education and the best schooling in Dhaka, the majority of its pupils and teachers are non-Christians. Despite its limitation, NDC can work as an effective model institution to expose and increase awareness of societal inequalities, particularly the society that fails to notice injustices and repressive economic structures that stratify society and suppress people.

### 4.3. Tuition: A Concern for Social Justice and Inclusive Education

While Catholic Church has a policy that no Catholic child would be refused a Catholic education due to a family's actual inability to pay fees, many students coming from low-income Catholic families cannot get enrolment at the college. However, preferential for the poor is an essential component in Catholic educational institutions (Canavan 2009; Gutiérrez 2009). While public primary and secondary schools in Bangladesh receive full government funding, private educational institutions do not. Many non-government schools and colleges get MPO (Monthly Pay Order) for their teachers. Notre Dame College used to get MPO for some of its teachers, but the college is no longer taking MPO. Since the college covers significant expenditures to provide quality education, therefore, the tuition fees are expensive and out of reach for students from low-income households. The majority of students come from medium to high-income families because NDC charges high tuition. Students from low-income families can only attend NDC if they qualify for tuition scholarships or financial grants, which are only available for Christian students (Gillespie and Peixotto 2001). The opportunities between the educated 'haves' and 'have-nots' grow more troublesome as the social distinction between them expands. While universal education is supposed to bring society closer together, this is rarely the case for developing nations like Bangladesh. In reality, education has become an institutionalized instrument for separating the rich from the poor (Khan et al. 2014). This, regrettably, also applies to NDC, whose reputed high-quality education is only marginally accessible to the poor and underprivileged groups.

### 5. Conclusions

This research reveals that the Catholic concept of social justice, social teachings and inclusive education are applied partially at NDC. In this scenario, there is a significant difference between intention and reality. Although NDC provides educational opportunities for the underprivileged, NDC is one of the city's most exclusive and strictly selective colleges, offering a privileged choice to the wealthy and well-connected. Furthermore, the institution fails to include students with disabilities because there is no facility at the college

for students with specific educational requirements. Moreover, the admission policy at the college is based on the performance of previous examinations; therefore, there is very little scope for the poor to get admitted to the college in regular educational programs. Moreover, it can be also argued that it is how the concepts 'poor' and 'preferential option' are being practiced at college that charges tuition and educates wealthy students. Studies claimed that Catholic schools around the world charge tuition and educate wealthy students (Grace and O'Keefe 2007). Researchers have noted that there are still discrepancies between declared commitments to social justice and how these commitments are really carried out in Catholic schools all around the world (Grace 2003; Grace and O'Keefe 2007). Moreover, Catholic education is a global education enterprise of the Catholic Church (Whittle 2015). As a result of this market economy, Grace (2002) argues that Catholic educators are adopting a policy of giving more priority to academic achievement sacrificing, the education missions, such as the preferential option for the poor. Notre Dame College in Bangladesh is not an exception to market economy policy as the college compromised its educational missions and preferential options for poor and underprivileged with academic excellence that exclude the poor and students with special educational needs from formal and regular educational programs. However, the college is also confronted with the problem of the nation's educational system that divides the rich and the poor.

**Funding:** The research was funded by Centennial Research Grant, Dhaka University. APC no: 807581fe45436c8f, Type: Invited.

**Institutional Review Board Statement:** This study was approved by HREC of the University of Hong Kong. The interviews were conducted for the author's PhD degree at HKU. Ethical approval No. EA1511012.

**Informed Consent Statement:** Informed consent was obtained from all subjects involved in the study.

**Conflicts of Interest:** The authors declare no conflict of interest.

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
