# Peer review of "Social Justice and Inclusive Education in Holy Cross Education in Bangladesh: The Case of Notre Dame College"

_religions, doi:10.3390/rel13100980_

Round 1

Reviewer 1 Report

The article highlights the problem of inclusivity in Catholic educational institutions in Bangladesh using the example of College Notre Dame, which is one of the schools managed by the Congregation of Holy Cross in the country. After reviewing the literature on the principals of inclusive education as understood by the Catholic Church and explaining the methodological issues, the author points out the basic issues related to the functioning of his chosen school such as the practical implementation of the option for the poor, the principle of social justice in the practice of school functioning, and the issue of tuition fees and its impact on the accessibility of the school. Having analysed these issues, the author of the article points out the necessary changes that are able to make Catholic schools more inclusive, in line with theoretical assumptions. An undoubted value and merit of the article is that it points out the implementation of the general principles that the Church sets out for its educational institutions to the challenges of a specific country or even a specific institution (college). It can also be argued that the specific dilemmas described on the example of a selected school have a more universal character. Of especial importance is the evaluation of the practical guidance of the preferential option for the poor, which is one of the basic principles of the social teaching of the Catholic Church. 

The article is clear, logically structured and properly divided into parts. It can also be considered relevant to educational theory, especially that related to Catholic schools. As already pointed out, the selected case study can help to raise questions and seek solutions elsewhere in the world as well. However, we do not learn from the content of the article whether it is needed and whether it answers the knowledge gap discovered. Looking at the cited publications, there is a certain lack of content with regard to recent publications (only 4 of the 52 cited sources are younger than 5 years). The teaching of the Catholic Church regarding the functioning of Catholic schools has also been treated too narrowly. As an example, the 2020 edition of the Directory for Catechesis, which devotes several points (309-312) to Catholic schools and contains interesting reflections also in the context of inclusivity, and unfortunately was not included in the reviewed article. The findings and conclusions are coherent and follow from both the research and the sources cited by the author. However, the conclusions are postulative in nature: "Holy Cross people at NDC need to respond to the call of their vocation to serve all people equally and to uphold a culture of mutual support and ownership for addressing the needs of all students." (lines 456-457); "NDC needs to evaluate its priorities" (lines 465-466), they lack indications of how to deal with the challenges and address the shortcomings associated with the demands for inclusivity in education.

The case study method adopted makes the research apparatus unclear, we do not know how many people took part in the interviews, what positions they held in the school, was there a standardised form used? As a result, we are dealing with opinions rather than measurable and verifiable data.

The article also adopts a rather narrow meaning of inclusivity. Nowadays there is a lot of focus on those who identify as sexual minorities, who have difficulties defining their own identity, also on those with special educational needs (e.g. on the autism spectrum), it would be interesting to extend to these or similar issues the consideration of inclusivity especially in relation to Catholic education, where one can see a certain tension between respect for every human being, especially those more in need of help and care, and fidelity to the teaching of the Magisterium and to an anthropology that is consistent with Revelation. Even if this is not a visible problem in Bangladesh (is it really not?) mentioning it in the first, theoretical part of the article, seems necessary.

Author Response

Point 1. I added more recent publications and reviewed the recommended document and integrated into the review.  

Point 2. I rewrite my conclusion based on the findings of my study.

Point 3. I included more information in the methodology section regarding the participant and their number and how I analyzed the data.

Point 4. I also addressed the opportunities for children with special educational needs at the college. I reviewed the issue in the literature review section.

Reviewer 2 Report

This is an excellent paper that juxtaposes Catholic education's mission and praxis in the context of the global south. The authors do a great job of presenting their arguments and supporting them with existing literature and original research. A few suggestions:

1. Provide a bit more context and situate your case study in the broader education system of Bangladesh. Some data on Bangladesh's high school/college education, number of SSC graduates seeking admission in colleges, number of applicants each year at NDC and acceptance rate, a visual of their acceptance rules and regulations, etc. would only add value to the article. There is an error in information on line 431 that states, "While public primary and secondary schools in Bangladesh receive full government  funding, private educational institutions do not." At least 98% of Bangladesh's secondary schools are privately managed with limited financial support (via MPO) from the government, not full funding.

2. Line 15, you may have misused the word "meritocracy." It has very particular meaning is education contexts (aka the myth of meritocracy). You may replace it with academic performance based admission...

3. You may want to improve your methodology section as well as research data presentation techniques. Please tell your readers how many interviews, for how long, how did you access the data, how you processed it, and if there was any trouble collecting the data. Also, you barely "use" your data. You have used less than five direct quotations from your interviews very casually. It would be good to know and help develop trust in your readers about the validity of your research if you tell us how many interviews, who the interviewees are, etc.

Author Response

Point 1. I added more information in the introduction, particularly about the case NDC.

Point 2. I included more information in the methodology section regarding the participant and their number and how I analyzed the data and added more quotations.

Point 3. I also corrected the information and changed the word meritocracy and replaced it with recommended words.

Reviewer 3 Report

The first paragraph of section 4.3 should be reviewed and edited. Some copy editing needs to be done elsewhere. The conclusion needs to be less prescriptive ("should") and based more on actual findings with clearer relationship between specific conclusions and particular warrant for the same. I wish there were fewer citations from the USA and more global (European, Latin American, African, certainly more Asian and beyond South Asia). The citations do not appear to follow Chicago style. For example, footnotes should not begin with the last name of the author but rather the first name. This is a solid piece and merits publication once revisions are made.

Author Response

Point 1. I did copy editing

Point 2. I added more recent publications from Asia and Africa and reviewed and integrated them into the review.  

Point 3. I rewrite my conclusion based on the findings of my study.